# Records and Occupation Time Statistics for Area-Preserving Maps [note 1]

**DOI:** 10.3390/e25020269

**Published:** 2023-02-01

**Authors:** Roberto Artuso, Tulio M. de Oliveira, Cesar Manchein

**Affiliations:** 1Dipartimento di Scienza e Alta Tecnologia and Center for Nonlinear and Complex Systems, Via Valleggio 11, 22100 Como, Italy; 2I.N.F.N, Sezione di Milano, Via Celoria 16, 20133 Milano, Italy; 3Departamento de Física, Universidade do Estado de Santa Catarina, Joinville 89219-710, SC, Brazil

**Keywords:** area-preserving maps, record statistics, infinite ergodicity

## Abstract

A relevant problem in dynamics is to characterize how deterministic systems may exhibit features typically associated with stochastic processes. A widely studied example is the study of (normal or anomalous) transport properties for deterministic systems on non-compact phase space. We consider here two examples of area-preserving maps: the Chirikov–Taylor standard map and the Casati–Prosen triangle map, and we investigate transport properties, records statistics, and occupation time statistics. Our results confirm and expand known results for the standard map: when a chaotic sea is present, transport is diffusive, and records statistics and the fraction of occupation time in the positive half-axis reproduce the laws for simple symmetric random walks. In the case of the triangle map, we retrieve the previously observed anomalous transport, and we show that records statistics exhibit similar anomalies. When we investigate occupation time statistics and persistence probabilities, our numerical experiments are compatible with a generalized arcsine law and transient behavior of the dynamics.

## 1. Introduction

One of the most remarkable advances in modern dynamics lies in the recognition that deterministic systems may exhibit statistical properties typical of purely stochastic processes: for instance, such systems may display diffusion properties similar to random walks [1,2,3,4]. Area-preserving maps (see, for instance, [1]) represent a prominent example of Hamiltonian systems where subtle features of dynamics, such as integrability vs. chaotic properties, may be studied. In this context, one of the most outstanding examples is represented by the Chirikov–Taylor standard map (SM) (see [1,5] and references therein). Even if we are only concerned with classical features, we recall that such a map played an important role in the development of quantum chaos: in particular, early investigations showed how quantum interference suppresses classical diffusion, a phenomenon called quantum dynamical localization [6]. Though the SM has been extensively explored by numerical simulations, very few rigorous results have been proven (see, for instance, the introduction in [7]): however, it is generally believed that for large nonlinearity parameters, this map typically exhibits good stochastic properties and sensitive dependence upon initial conditions. Here a remark is due: such a map can be studied either on a 2-torus or an (unbounded) cylinder: the latter representation is naturally adopted when transport properties are concerned, and analogies with random walks are taken into account [1,3,8,9]. While particular nonlinear parameters in the standard map can be tuned to generate strong anomalous diffusion [10] (see also [11,12,13,14,15]), here we will only deal with the case in which diffusion is normal. Our findings will be compared with those obtained for another area-preserving map, characterized by the lack of exponential instability: the so-called Casati–Prosen triangle map (TM) [16], introduced by considering, in an appropriate limit, the Birkhoff dynamics of a triangular billiard: apart from its intrinsic interest, such a map is an ideal benchmark to test whether stochasticity properties, exhibited by strongly chaotic systems, are also showcased by systems lacking any exponential instability. It also turns out that many features of the TM are still debated, starting from basic properties, such as ergodicity and mixing (see, for instance, [17,18]). It is also worth mentioning that the TM has been studied in the quantum setting to investigate which features of quantum chaos are displayed in the absence of classical exponential instability [19,20].

In detail, we will compare different indicators for both maps on the cylinder: though, in principle, further complications are added when one considers a non-compact phase space [21,22], this is the appropriate scenario to discuss transport properties and record statistics and to check whether tools from infinite ergodic theory may enrich our understanding of such systems.

Our main findings are that the SM, in its typical chaotic regime, displays all stochastic properties of a purely stochastic system, while, as expected, results are far more complicated for the TM; we believe that some new insight is provided by our analysis, in particular, as regards persistence behavior, occupation time statistics, and the relationship between transport properties and record statistics. When we compare the records statistics for the SM and the TM, the latter exhibits an anomalous growth, which can be related to transport properties. The difference is even more striking when we consider occupation time statistics and survival probability: the SM again behaves like a simple random walk, while the TM numerical experiments suggest transient behavior, coexisting with a generalized arcsine law.

The paper is organized as follows. In Section 2, the Chirikov–Taylor standard map (Equation 1) and the triangle map (Equation 3), our basic models, are presented, and we also mention the main properties we analyze. Section 3 is dedicated to discussing transport properties, records statistics, and occupation time statistics. We end with a discussion in Section 4.

## 2. The Basic Setting

We recall the definition of the SM
(1)$$\begin{array}{c} p_{n+1}=p_{n}+\frac{K}{2\pi }\sin\left(2\pi x_{n}\right),\\ x_{n+1}=x_{n}+p_{n+1}\;\mathrm{mod}\;1; \end{array}$$
where *x* and *p* are canonical variables: (Equation 1) is the stroboscopic map generated by the time-dependent Hamiltonian $H\left(p,x,t\right)=p^{2}/2+K/{\left(2\pi \right)}^{2}\cos\left(2\pi x\right)\sum_{m=-\infty }^{+\infty }\delta \left(t-m\right)$, describing a periodically kicked rotator, *K* being the nonlinear parameter: when *K* is sufficiently big, no KAM invariant circles bound the motion, and one can study moments of the diffusing variable p∈R:(2)$$\langle {\left|p_{n}-p_{0}\right|}^{q}\rangle ∼n^{q\nu (q)}.$$

The typical behavior observed for the second moment in simulations is normal diffusion ν(2)=1/2 [23,24], while, for certain parameter values, the existence of stable running orbits (accelerator modes) induces superdiffusion, ν(2)>1/2) [11,12,13,14,15,25,26,27]. We point out that a finer description of anomalous transport is obtained by considering the full spectrum ν(q): if ν(q)=α·q, for some α≠1/2 one speaks about weak anomalous diffusion, whereas the case of a nontrivial ν(q) is dubbed strong anomalous diffusion [10]. As far as the SM is concerned, we will consider the case where transport in the stochastic sea is normal, even if the phase space exhibits a mixture of chaotic and elliptic components (see Figure 1). We will come back to this point in the final discussion.

On the other side, the TM is defined (on the cylinder) as:(3)$$\begin{array}{c} p_{n+1}=p_{n}+2\left(x_{n}-⇂x_{n}↿-\mu {\left(-1\right)}^{⇂x_{n}↿}\right),\\ x_{n+1}=x_{n}-2p_{n+1}\;\mathrm{mod}\;2, \end{array}$$
where ⇂⋯↿ denotes the nearest integer. It was introduced in [16] (see also [28]) as an approximate Birkhoff map of irrational triangular billiards: systems lacking exponential instability, whose ergodic properties are subtly related to irrationality properties of the angles [29,30,31,32]: in particular, the original derivation refers to an elongated triangular billiard, with a very small angle: *x* and *p* are Birkhoff coordinates describing successive collisions with the small side, and the parameter μ is associated with the asymmetry between the two non-small angles. Figure 2 shows the function $\delta p\left(x\right)=p_{n+1}-p_{n}=2\left(x_{n}-⇂x_{n}↿-\mu {\left(-1\right)}^{⇂x_{n}↿}\right)$. Since the derivative of δp(x) (apart from discontinuity points, where it does not exist) is constant $\delta^{\prime }\left(x\right)=2$, the jacobian of the map is
(4)$$J\left(x,p\right)=J=\left(\begin{array}{cc} -3 & -2\\ 2 & 1 \end{array}\right).$$

The determinant is 1 (the map is area-preserving, while the trace is −2, corresponding to double degenerate eigenvalues λ=−1. This implies that the TM lacks any form of exponential instability.

We remark that the polygonal billiards represent both a hard mathematical challenge [33,34,35,36] and a natural benchmark when trying to assess which microscopic dynamical features lead to macroscopic transport laws [37,38,39] (see also [40,41]). In this respect, it is worth mentioning that anomalous transport has been associated with scaling exponents of the spectral measure [42] and that generalized triangle maps have been investigated recently, both as connected to dynamical localization [43] and with respect to slow diffusion [44]. A typical phase portrait (on the torus) of the TM is shown in Figure 3.

Before mentioning the numerical experiments we performed, a crucial observation is in order. When looking at transport properties (and records statistics), considering maps on the cylinder is quite natural, while from the ergodic point of view, this perspective is somehow delicate since no normalizable invariant density exists [21,22], and the appropriate setting is an infinite ergodic theory (we will not employ the full machinery of infinite ergodic theory; however, the lack of an invariant probability measure is essential when properties such as occupation time statistics are examined). When polygonal channels are considered, even establishing recurrent properties of the dynamics is a demanding task [45].

The first set of properties we investigated is more conventional, and a few results, as we will mention in the next section, have already been considered, especially as far as the SM is concerned. We will look at transport properties, in particular through the first and the second moment of the diffusing variable. We will also study records statistics, which recently have become very popular (see [46,47] and references therein). Then we will scrutinize statistical properties, such as persistence probability and (generalized) arcsine law [48,49]. While motion in the stochastic sea for the SM will exhibit typical properties of a simple stochastic process like a random walk, our findings for the TM suggest both the existence of transient dynamics and anomalous scaling exponents for records statistics and generalized arcsine law.

## 3. Results

We start by considering properties associated with the spreading of trajectories over the phase space; then we will consider occupation time statistics.

### 3.1. Diffusion

This is a warm-up exercise since transport properties have been studied both for the SM [1,23,24] and for the TM [32]. We observe *normal* transport for the case of the SM (see panels (a) and (b) in Figure 4, where $\langle \left(p_{n}-p_{0}\right)\rangle ∼n^{1/2}$ and $\langle {\left(p_{n}-p_{0}\right)}^{2}\rangle ∼n$, respectively), while the TM results indicate a superdiffusion (see panels (a) and (b) in Figure 5), with
(5)$$\langle {\left(p_{n}-p_{0}\right)}^{2}\rangle ∼n^{1.86},$$
in agreement with [32]. We remark that by looking at the power-law exponents of the first two moments, we find that anomalous diffusion is possibly weak [10], namely if we consider the full spectrum of moments’ asymptotics:(6)$$\langle {\left|p_{n}-p_{0}\right|}^{q}\rangle ∼n^{\phi (q)},$$
we have a single scaling, in the sense that
(7)ϕ(q)=α·q;
where normal diffusion is recovered when α=1/2. This is reasonable since weak anomalous diffusion has been observed in polygonal billiards [50].

### 3.2. Average Number of Records

The statistics of records are very popular in the analysis of correlated and uncorrelated stochastic time sequences [46,47]. Since this subject has not been explored thoroughly in the deterministic setting (with the remarkable exception of [51,52]), we briefly review the basic concepts.

First of all, let us recall the (straightforward) definition of a record: given a sequence of real data $x_{0},\;x_{1},\;\cdots ,x_{k},\;\cdots$ the element $x_{m}$ is a record if
(8)$$x_{m}>x_{j}\;j=0,\;1,\;\cdots m-1,$$
(we consider $x_{0}$ to be the first record). To the sequence of data points we associate the binary string $\sigma_{0},\;\sigma_{1},\;\cdots ,\sigma_{k},\;\cdots$, where
(9)$$\sigma_{l}=\left\{\begin{array}{c} 1\;\mathrm{if}\;x_{l}\;\mathrm{is}\;a\;\mathrm{record}\\ 0\;\mathrm{otherwise} \end{array}\right.$$

The number of records up to time *N* is then
(10)$$M_{N}=\sum_{j=0}^{N}\;\sigma_{j}.$$

The properties of the average number of records, $\langle M_{N}\rangle$, and the corresponding variance
(11)$$Var\left(M_{N}\right)=\langle M_{N}^{2}\rangle -{\langle M_{N}\rangle }^{2}$$
are important tools to access the nature of the data sequence: as a matter of fact, if the different $x_{j}$ are independent identically distributed random variables, then, for large *N*, we have [53,54]:(12)$$\langle M_{N}\rangle =\ln N+\gamma_{E}+O\left(N^{-1}\right),$$
where $\gamma_{E}=0.5772\cdots$ is the Euler–Mascheroni constant, and
(13)$$Var\left(M_{N}\right)=\sigma^{2}\left(N\right)=\ln N+\gamma_{E}-\frac{\pi^{2}}{6}+O\left(N^{-1}\right).$$

We remark that both quantities are independent of the common distribution of the random variables: this universality is an important feature of record statistics in different contexts.

The results are quite different for a correlated sequence, as when $x_{j}$ denotes the position of a random walker at time *j*:(14)$$x_{j+1}=x_{j}+\xi_{j+1},$$
where the jumps are taken from a common distribution ℘(ξ). In this case, the behavior is [46,47]:(15)$$\langle M_{N}\rangle \approx \frac{2}{\sqrt{\pi }}\sqrt{N},$$
and
(16)$$Var\left(M_{N}\right)\approx 2\left(1-\frac{2}{\pi }\right)N,$$
so that the standard deviation is of the same order of magnitude as the average. Again this is a *universal* result, independent of the particular jump distribution ℘(ξ), as long as the distribution is continuous and symmetric. The crucial ingredient of the proof is that the process renews as soon as a new record is achieved and the appearance of the new record is related to the survival probability for the process, which is universal in view of the Sparre–Andersen theorem [49,55,56] (see also [57]).

Numerical results on records statistics are reported in Figure 4 (for the SM) and Figure 5 (for the TM), panels (c) and (d) for the average number of records 〈M(n)〉 and variance $\sigma^{2}\left(n\right)$, respectively. For the SM, our results plotted in Figure 4 are consistent with early investigations [51,52] and with the asymptotic behavior of a random walk: our fits are in excellent agreement with $\langle M\left(n\right)\rangle ∼n^{1/2}$ and $\sigma^{2}\left(n\right)∼n$, respectively. For the TM, we observe anomalous scaling with regard to (Equation 15) and (Equation 16), as plotted in Figure 5: the behavior is related to the transport properties in the sense that data are consistent with the growths.
(17)$$\langle M_{N}\rangle ∼N^{\phi (1)},\;Var\left(M_{N}\right)∼N^{\phi (2)}.$$

Similar behavior was observed in [51,52] for the SM in the presence of accelerator modes. We remark that, though in the following we will fix our attention to a particular parameter value for the TM, we checked that reported experiments do not depend on the particular parameter choice, as exemplified in Figure 6, where the growth of the averaged number of records is reported for three different parameters of the TM: $\mu =\sqrt{7},\sqrt{2}/2,$ and $\mu =(\sqrt{5}+e)/12$ plotted with dashed-dotted (green), dashed-dotted-dotted (magenta), and dashed (red) lines. While the actual choice of μ parameter values is somehow arbitrary, the only provision is to choose irrational values. Otherwise, the TM becomes pseudointegrable [32,33,34]. Figure 6 shows that the asymptotic power-law fitting parameters are almost the same (a∼0.15 and γ∼0.9) for different irrational values of μ.

We also remark that the number of initial conditions and iteration time plays a role as regards the simulations we will present later on: while our choices are gauged by the stability of the plots upon variations in this parameter, survival probability simulations require extremely large iteration times, as remarked in the caption of Figure 7.

While a general, quantitative relationship (if any) between transport exponents and statistical properties of records has not been fully developed, to the best of our knowledge, it is possible in some cases to connect ϕ(1) to the expected maximum of the walk [58,59], that, for a random walk with unit jumps, coincides with the number of records. On the other side, we mention that non-homogeneous random walks offer examples where such a relationship does not hold [60,61,62,63,64].

### 3.3. Occupation Time Statistics

When we consider the evolution of the cylinder, both for the SM and the TM, we are in the presence of infinitely ergodic systems [21,22]. Since, while the Lebesgue measure is preserved, due to area conservation, the (constant) phase space is unbounded, so the invariant density cannot be normalized. This has a series of remarkable consequences, which originally have been considered in the context of stochastic processes, and then explored in the deterministic evolution framework.

One of the most striking properties that has been investigated is the generalized arcsine law (see [48] for the standard formulation for stochastic processes): we briefly recall the main result that lies at the basis of our analysis, namely Lamperti’s theorem [65]. The original formulation involves discrete stochastic processes, for which the infinite set of possible states can be separated into two sets, *A* and *B*, separated by a single site $x_{0}$, such that a transition from one set to the other can only be achieved by passing through $x_{0}$, which can be taken as the starting site, and is supposed to be recurrent (namely the probability of returning to it is 1). For instance, we can think of a one-dimensional random walk on an integer lattice, with $x_{0}=0$ and *A* (*B*) consisting of strictly positive (negative) lattice sites. We are interested in the limiting distribution of N(n)/n, the fraction of time spent in the positive semi-axis up to time *n*. The theorem states that such a distribution exists in the n→∞ limit, and it is characterized by two parameters α and η. η is related to the symmetry properties of the process, being the expectation value of the fraction of time spent in $R_{+}$:(18)$$\eta =\lim_{n\to \infty }E\left(\frac{N(n)}{n}\right):$$
for a symmetric process η=1/2, and from now on, we will only consider such a case.

The other parameter, α, is instead connected to the behavior of the generating function of first return probabilities to the starting site: it can be shown [66] that it can be related to the probability $P_{n}$ of being at the starting site after *n* steps in the following way.
(19)$$P_{n}∼\frac{H(n)}{n^{1-\alpha }},$$
where H(n) is a slowly varying function, namely
(20)$$\lim_{n\to \infty }\frac{H(yn)}{n}=1.$$

Under such conditions, the density of φ=N(n)/n in the infinite time limit is given by Lamperti distribution:(21)$$G_{\alpha }\left(\varphi \right)=\frac{\sin(\pi \alpha )}{\pi }\frac{\varphi^{1-\alpha }{\left(1-\varphi \right)}^{1-\alpha }}{\varphi^{2\alpha }+2\varphi^{\alpha }{\left(1-\varphi \right)}^{\alpha }\cos\left(\pi \alpha \right)+{\left(1-\varphi \right)}^{2\alpha }},$$
which reproduces the usual arcsine law
(22)$$P\left((N_{n}/n)\leq \xi \right)=\frac{2}{\pi }\arcsin\left(\sqrt{\xi }\right)$$
when α=1/2, in the universality class of the Sparre–Andersen theorem. Deviations from standard arcsine law have been reported in several cases in the framework of deterministic dynamics [67,68,69,70,71,72,73,74], mainly in the context of intermittent maps. Numerical experiments for the SM confirm the validity of the arcsine law, α=1/2, see panel (a) in Figure 8. To the best of our knowledge, this is the first time such an indicator has been considered in the analysis of area-preserving maps.

The results, as expected, are quite different for the TM, and they suggest novel features exhibited by this map. In particular (see panel (b) in Figure 8), numerical results are well fitted by a Lamperti distribution (with α≈0.42) and thus are different from an ordinary random walk), except for those that present enhanced peaks. Intuitively, such an additional contribution might be due to a fraction of orbits never returning to the origin: this would correspond, in stochastic language, to a transient random walk (we recall that according to Pólya’s theorem [75] a simple symmetric random walk is recurrent (so the return to the starting site is sure) in one and two dimensions, and transient in higher dimensions). Such a possibility is indeed not excluded for infinite polygonal channels [45].

Our last set of simulations concerns the survival probability [68]:(23)$$P_{cum}\left(n\right)=\mathrm{prob}\left(p_{n}\geq 0\cdots p_{1}\geq 0|p_{0}=0\right).$$

When considering recurrent random walks, the asymptotic behavior of the survival probability is again ruled by the Lamperti exponent [65,66] (see also [76]):(24)$$P_{cum}\left(n\right)∼n^{-\alpha }.$$

Once again, SM simulations (see panel (a) in Figure 7) agree with the expected behavior for simple random walks (α=1/2), while the situation is completely different for the TM, where the survival probability seems to tend to a finite limit for large *n*, see panel (b) in Figure 7. This is consistent with the transient nature of the TM, which we conjectured in the analysis of generalized arcsine law.

## 4. Discussion

We have performed a set of extensive numerical experiments on two paradigmatic area-preserving maps, the SM and the TM, focusing on the case where such maps are considered on a cylinder, namely a non-compact phase space. First, we reproduced known results about normal diffusion for typical (chaotic) parameters of the SM and superdiffusion for the TM. In particular, for the TM, we recover the anomalous spreading exponent reported in [32]. Then we explored records statistics: numerical simulations again confirm that the SM behaves like a simple random walk in accordance with simulations in [51,52], while anomalous growth is exhibited by the TM. We remark that, in both cases, the exponent that accounts for the growth of the number of records is quite close to the one that determines the asymptotic behavior of the first moment of the transporting variable, see (Equation 17). This is also consistent with the results reported in [51,52] for the SM in the presence of accelerator modes. While we only considered the case for which the SM displays normal diffusion (up to the time scale we were able to investigate), it might be possible that accelerator modes (of the possibly very high period) appear for almost every nonlinear parameter *K* (see the discussion in [12]). This is an issue of conceptual relevance since accelerator modes might eventually determine the true asymptotic behavior, but, on the other side, the time scale on which such behavior is exhibited might be beyond any possible numerical simulation. The most interesting results arise in the analysis of occupation times, such as generalized arcsine law and survival probability: to the best of our knowledge, such properties are investigated here for the first time in the context of area-preserving maps. For the SM, we recover the usual arcsine law and a survival probability in the Sparre–Andersen universality class, while the TM displays different behaviors. In particular, the distribution of occupation times in the positive half-axis (for the transporting variable) is well-fitted by a superposition of a generalized Lamperti distribution (with a different exponent with regard to the SM) and a sum of two δ peaks at the extreme values, which we ascribe to transient orbits. We stress that the analysis of survival probabilities supports such a conjecture since they seem to attain a non-zero limit for very long times. The development of further tools to sustain such a picture is a line of research that we hope will be pursued in further studies, which might also lead to new stochastic modeling of the TM [44]. From a different perspective, it would also be important to consider classes of area-preserving maps for which an analytic approach is feasible. A remarkable example in this respect is provided by the Cerbelli–Giona map [77,78], where jumps in momentum are given by a tent map. Such a map probes the effect of singularities (that are also present in the TM, though without any exponential instability). In particular, in [78], an exact computation of the diffusion coefficient is presented, which bears some similarities to analogous computations for one-dimensional maps [79,80].

## Figures and Tables

**Figure 1 entropy-25-00269-f001:**
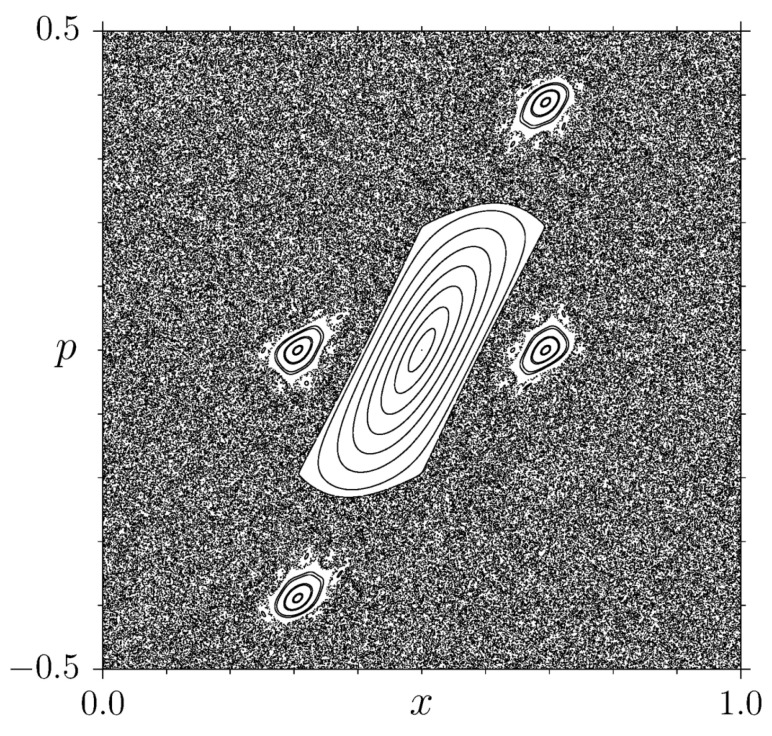
Phase-space portrait for the standard map (Equation 1) on the 2-torus, for K=2.6. Here 40 uniformly distributed initial conditions were used for *x*, while maintaining $p_{0}=0$ fixed: each initial condition is iterated $10^{4}$ times.

**Figure 2 entropy-25-00269-f002:**
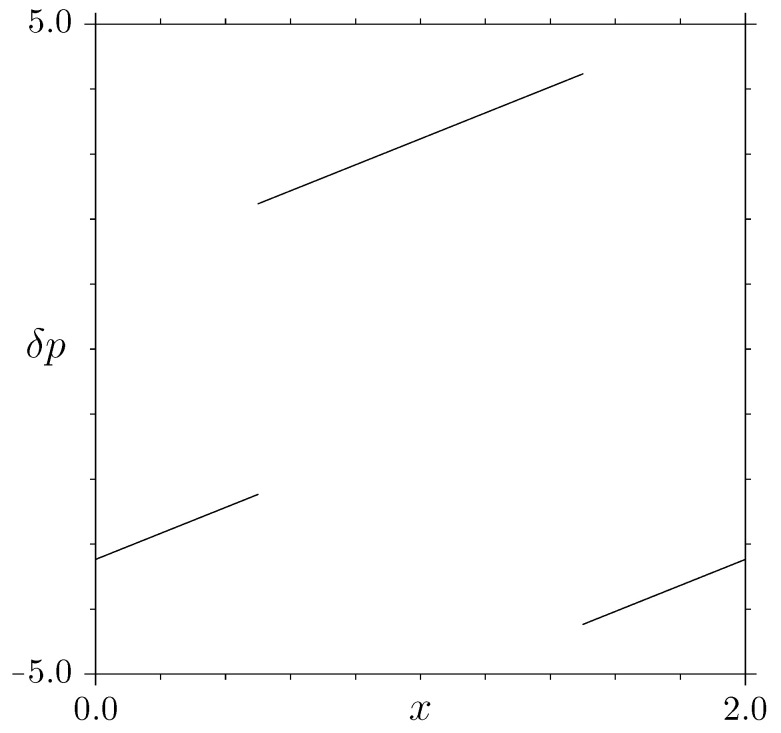
$\delta p=p_{n+1}-p_{n}$ as a function of *x* for the triangle map (Equation 3), for $\mu =\frac{1+\sqrt{5}}{2}$ (golden mean).

**Figure 3 entropy-25-00269-f003:**
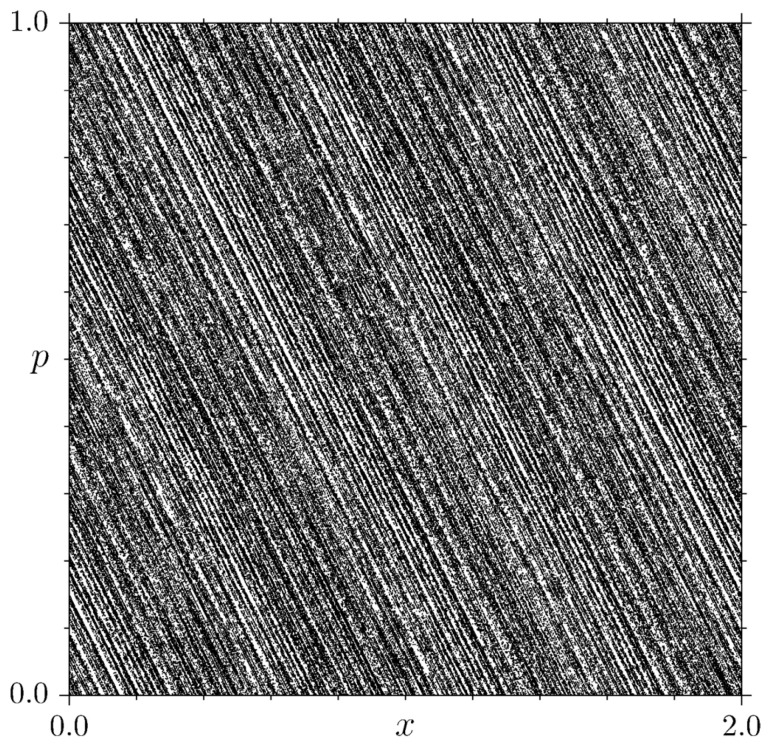
Phase-space dynamics for the triangle map (Equation 3), for $\mu =\frac{1+\sqrt{5}}{2}$ (golden mean). Here 100 randomly distributed initial conditions were used for *x* and *p*: each initial condition is iterated $5\times 10^{4}$ times. Notice the typical filament structure in the phase space [30,31].

**Figure 4 entropy-25-00269-f004:**
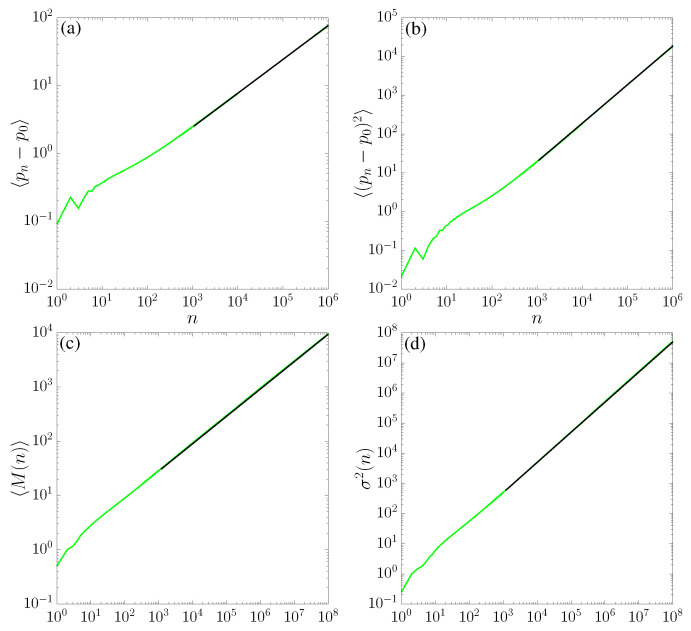
(**a**) Average number of records, (**b**) variance, (**c**) first, and (**b**) second moments of variable *p* for K=2.6 in the standard map (Equation 1), as a function of time. These quantities were computed for $10^{6}$ initial conditions for $x_{0}$, arbitrarily chosen in the chaotic sea along the line $p_{0}=0$. Black-continuous lines correspond to power-law asymptotics fit $F\left(n\right)=an^{\gamma }$: the fitting parameters are, for (**a**) a=0.77(7), γ=0.50(1), for (**b**) a=0.02(1), γ=0.99(1), for (**c**) a=0.86(0), γ=0.50(9), and for (**d**) a=0.50(7), γ=1.00(3).

**Figure 5 entropy-25-00269-f005:**
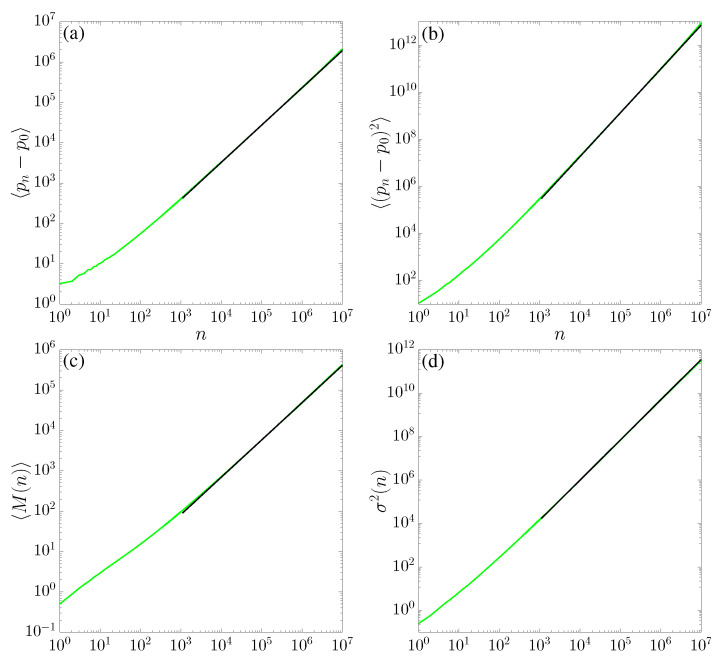
(**a**) Average number of records, (**b**) variance, (**c**) first, and (**b**) second moments of variable *p* for the golden mean $\mu =\frac{1+\sqrt{5}}{2}$ in the triangular map (Equation 3) as a function of time. These quantities were computed for $10^{6}$ initial conditions for $x_{0}$, arbitrarily chosen in phase space along the line $p_{0}=0$. Black-continuous lines correspond to power-law asymptotics $F\left(n\right)=an^{\gamma }$: the fitting parameters are, for (**a**) a=0.65(4), γ=0.92(4), for (**b**) a=0.67(6), γ=1.86(0), for (**c**) a=0.13(9), γ=0.92(4), and for (**d**) a=0.04(0), γ=1.84(9).

**Figure 6 entropy-25-00269-f006:**
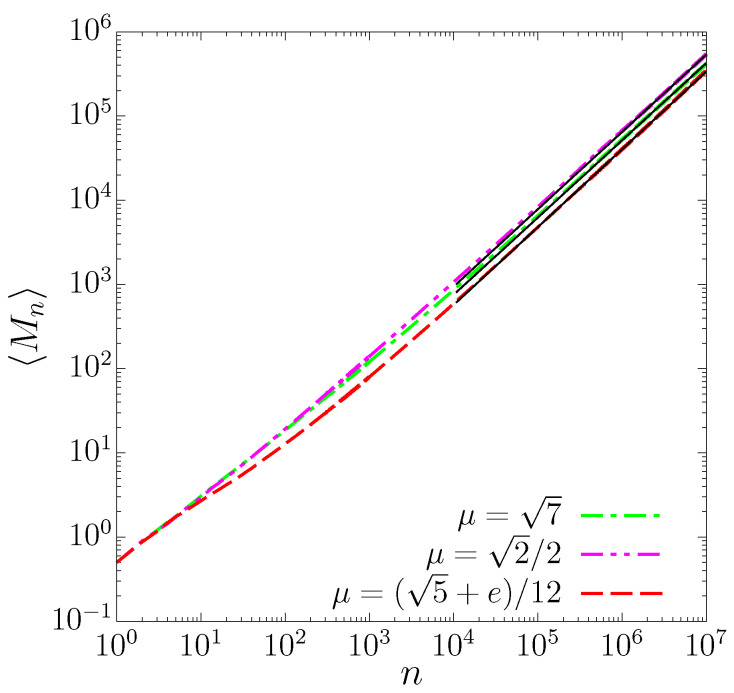
Average number of records for three additional parameters μ in the TM. These quantities were computed for $5\times 10^{5}$ initial conditions for $x_{0}$, arbitrarily chosen in phase space along the line $p_{0}=0$. Black-continuous lines correspond to the power-law asymptotic fitting function $F\left(n\right)=an^{\gamma }$, with (a,γ)=[0.20(1),0.91(8)] (magenta), (a,γ)=[0.16(2),0.91(7)] (green), and (a,γ)=[0.11(5),0.92(4)] (red).

**Figure 7 entropy-25-00269-f007:**
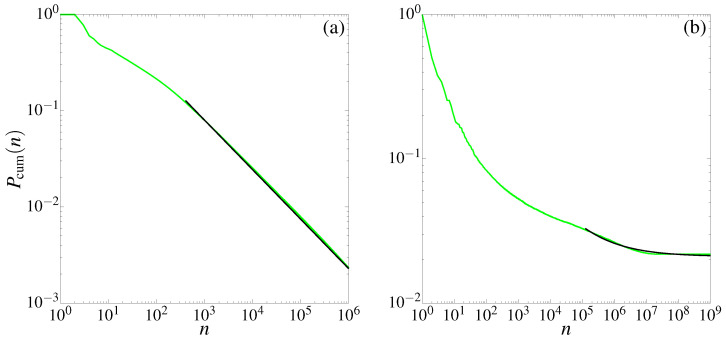
Cumulative distribution function for the survival times obtained for variable *p* for (**a**) the standard map (Equation 1) and (**b**) the triangle map (Equation 3) on a logarithmic scale. Data are obtained by simulating $10^{6}$ and $10^{5}$ initial conditions, respectively. Continuous black lines correspond to power-law asymptotic functions $F\left(n\right)=a+bn^{-\alpha }$: the fitting parameters are a=0,b=2.80(0), and α=0.51(5) in (**a**) and a=0.021(0),b=1.62(6), and α=0.42(0) in (**b**). Notice that to observe the asymptotic plateau for the TM, we have to go to very large iteration times.

**Figure 8 entropy-25-00269-f008:**
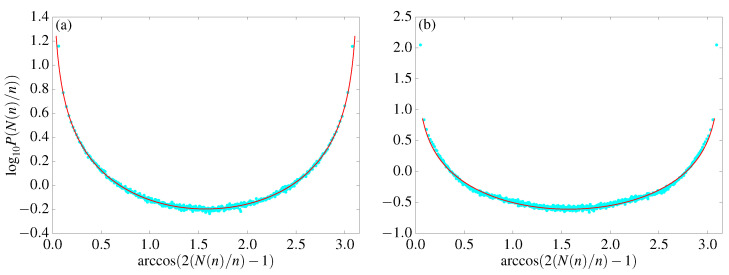
Distribution of the fraction of time spent in the positive axis for the momentum *p* in the standard (Equation 1) (**a**) and triangle (Equation 3) (**b**) maps on a semi-logarithmic scale. To enhance the readability of the border values, the transformation x→arccos(2x−1) on the horizontal axis. The (light blue) points represent the simulation results, and the (red) line is the Lamperti distribution (Equation 21). Data are obtained by computing $10^{6}$ initial conditions iterated $10^{6}$ times for the standard map and $10^{6}$ initial conditions iterated $10^{8}$ times for the triangle map. The fitting parameters are α=0.49(9) for (**a**) and α=0.42(0) for (**b**). In the case of the TM, data suggest a superposition of a (rescaled) Lamperti distribution and two Dirac’s δ centered on x=0 and x=1 (see text).

## Data Availability

Not applicable.

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
