# Peer review of "Records and Occupation Time Statistics for Area-Preserving Mapsâ€"

_entropy, 2023, doi:10.3390/e25020269_

Round 1
Reviewer 1 Report
please see attached

Reviewer 2 Report
Report on the "Records and occupation time statistics for area-preserving maps" manuscript for Entropy.
In this manuscript, the authors study transport properties in non-compact phase spaces of two area-preserving deterministic systems, namely the Chirikov-Taylor standard map and the Casati-Prosen triangle map. In more detail, they numerically investigate diffusion properties by measuring records’ statistics and occupation time statistics for ensembles of initial conditions chosen inside chaotic regions of their respective phase spaces. The main conclusion they draw is that the transport properties for the standard map exhibit simple random walk effects while for the Casati-Prosen triangle map these properties are rather different and resemble more transient dynamical properties.
General comments:
· The authors should put more effort to report early in the manuscript (starting maybe with the abstract) which are the main findings of their study. The repeatedly mention something along the lines “… our findings are quite different in the case of the triangle maps when compared to those for the standard map”. One has to reach the end of the results’ and discussion section to find that the considered models exhibit differences in e.g. in the respective parameters associated with rate of diffusion or asymptotic trends of the statistical measurement used in their work etc. In my opinion, the reader expects to get earlier a flavor or “what is different” and surprising in this study.
· In several parts of the manuscript, the language needs to be improved. See my detailed comments later.
Detailed comments:
1. The last sentence in the abstract is rather long and not so easy to read.
2. Lines 17-19: This sentence is not well written and clear.
3. In the introduction and the discussion of the standard map and anomalous diffusion, there are also other related studies that deserve discussion, see for example:
· R. Venegeroles, “Calculation of Superdiffusion for the Chirikov-Taylor Model”, Phys. Rev. Lett. 101 (5) 054102 (2008).
· Manos, Robnik “Survey on the role of accelerator modes for anomalous diffusion: the case of the standard map”, Phys. Rev. E, 89, 022905 (2014).
· M. Harsoula, G. Contopoulos, Global and local diffusion in the standard map, Phys. Rev. E 97 022215 (2018).
· M. Harsoula, K. Karamanos, G. Contopoulos, Characteristic times in the standard map, Phys. Rev. E 99 032203 (2019).
· Moges et al. “Anomalous diffusion in single and coupled standard maps with extensive chaotic phases”, Physica D: Nonlinear Phenomena, 431, 133120 (2022).
4. In the introduction and the discussion of the triangle map and its diffusion properties, there are also other related studies that deserve discussion, see for example:
· M. Degli Esposti et al. “A semi-classical study of the Casati-Prosen triangle map” Nonlinearity 18 1073 (2005).
· Jiaozi Wang et al. “Statistical and dynamical properties of quantum triangle map” J. Phys. A: Math. Theor. 55 234002 (2022).
5. Lines 42-46: This text consists of one single paragraph expressed in one only sentence. It is not well written making it difficult to understand its message.
6. A brief description regarding the variables and parameters in the two respective models in Section 2 is required.
7. Lines 76-78: minor suggestion regarding the repetition of the word “considered” in the sentence also used several times in the same paragraph.
8. Lines 80-83: are not so well written. Furthermore, when the authors say “.. while our findings are quite different in the case of the TM”, the reader expects to get a flavor or “what is different”. This is the third time, if I am not mistaken (abstract, + introduction + model discussion so far), that this finding is mentioned in a rather vague way. There is no mention so far of what is the main finding. On the other hand, this information (maybe not so surprising as it well documented in the literature) is provided immediately for the standard map.
9. Line 118: maybe the authors mean “consistent” instead of “coherent”?
10. The discussion of Figure 5 needs to mention explicitly which parameter in the model is considered. It is only mentioned in the caption. Why the author chose these particular values for Figure 5? What is the main motivation? Are these three values representative in some way? Moreover, it is not clear to me, which data (parameter) the fit function is fitting. Generally, speaking this figure is purely discussed in the text.
11. The authors should justify the rationale in considering different size of initial conditions and final interpretation times. For example, they seem to be different in Figure 4, 6 and 7 (I am not sure about Figure 5 – see also my previous comment). I suppose that for Figure 7(b) the authors needed to integrate for much longer iterations and the sample size was a computational issue.
12. The authors should expand the discussion of the results and make an effort to better highlight their new findings and their importance.
13. I find the Discussion section very short. I would suggest to make to better link the work done here with the existing literature and further elaborate on its importance and general future outlook.
In general, I believe this an interesting work overall and research direction. In my opinion, the manuscript at its current state, it is not ready for publication in Entropy. It requires much more work, extensive, and careful rewriting. If the authors amend the paper according to my detailed suggestions, then I will be gladly to evaluate it again. My overall recommendation is “major revision”.
Round 2
Reviewer 1 Report
The authors have addressed my points in the revised version. I recommend it for publication.
Reviewer 2 Report
The authors have ammended the manuscript taking into account all my comments and suggestions.